# Investigation of the Pull-Out Behaviour of Metal Threaded Inserts in Thermoplastic Fused-Layer Modelling (FLM) Components

Tobias Kastner *,†, Juliane Troschitz †, Christian Vogel, Thomas Behnisch, Maik Gude and Niels Modler

Institute of Lightweight Engineering and Polymer Technology, Technische Universität Dresden, Holbeinstraße 3, 01307 Dresden, Germany
* Correspondence: tobias.kastner@tu-dresden.de
† These authors contributed equally to this work.

**Abstract:** To provide detachable, secure and long-term stable joints in fused-layer modelling (FLM) components or assemblies, metal threaded inserts are widely used as extrinsic interfaces. However, the load-bearing capacity of such inserts is influenced by the inhomogeneous, anisotropic material structure of the FLM components. This work evaluates the influence of the joining zone design and the printing process parameters on the achievable joint properties. Therefore, we printed thermoplastic FLM test specimens with varying parameters for infill density, wall thickness, layer height and nozzle temperature. Subsequently, metal threaded inserts were warm-embedded into the test specimens and investigated in quasi-static pull-out tests. The results show that the infill density in the joining zone has the largest impact on the joint strength and should be 70% or higher. Furthermore, an analysis of different wall thicknesses around the pre hole shows that a minimum value of 2.4 mm is required for the selected insert geometry to achieve a high pull-out force. Increasing the wall thickness beyond this value does not significantly affect the joint strength. The results provide an improved base for detailed understanding and interface design in FLM components for the integration of metal threaded inserts as well as for further investigations regarding different printing materials and load types.

**Keywords:** additive manufacturing; fused-layer modelling; joining; metal threaded insert; pull-out test



## 1. Introduction

Since their invention in the 1980s, rapid prototyping or additive manufacturing (AM) technologies have developed rapidly and are currently gaining increasingly importance in the industrial sector [1–4]. Particularly in the area of individualized products, small series or spare parts, AM technologies are widely applied. In addition to a variety of AM technologies for different materials, various processes are available on the market specifically for polymers [5,6]. In addition to powder-based processes, extrusion-based technologies, such as fused-layer modelling (FLM), are of particular industrial relevance [7] since they require a relatively low investment in equipment and materials, process a wide range of materials and are considered robust in terms of process technology.

However, a key challenge in the industrial use of FLM components is their interface with other components and/or assemblies [8]. Therefore, classical joining technologies (e.g., bonding, welding or screwing) are available, as they are also used for injection moulded plastic components or continuous fibre-reinforced thermoplastics [9–13]. In order to enable detachable, secure and long-term stable joints with FLM components, the use of threaded inserts as a so-called extrinsic interface provides a suitable solution [14].

Typical applications are in the manufacturing of individual components for medical aids and consumer products or technical prototyping (e.g., for retrofitting parts and sensors

in autonomous driving (Figure 1). Thereby, a significant advantage is the possibility to easily detach the joint with good suitability for repeat assembly [15]. This allows the replacement of individual components in the event of failure without having to completely re-manufacture the complex assembly [2]. In addition, metal inserts can significantly increase the load-bearing capacity of bolted joints by reducing stress concentrations around the hole [16].

A number of established different embedding technologies exist for integrating metal threaded inserts into thermoplastic components [15,17], including warm and ultrasonic embedding. Omidvarkarjan et al. [18] presented an insert for the integration of female threads into AM polymer parts, which consists of a metal threaded insert, an AM optimized female cutout and an AM polymer clip.

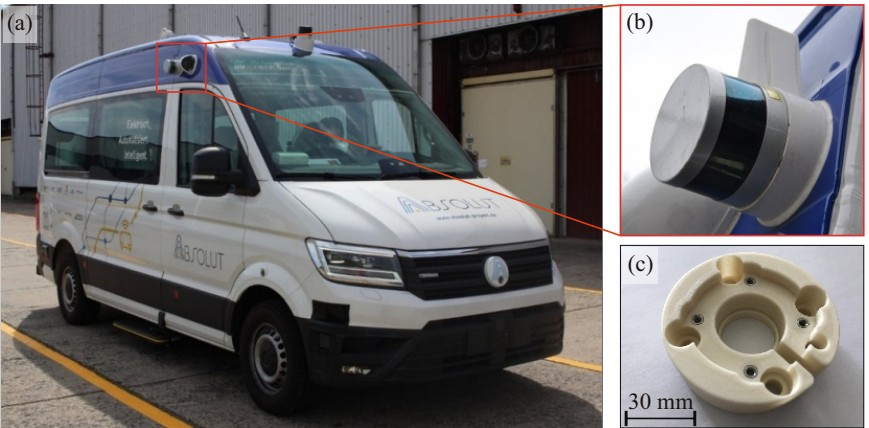

**Figure 1.** Demonstrating the use of metal threaded inserts in FLM components for retrofitting sensors for autonomous driving within the ABSOLUT project: (**a**) automated electric VW e-Crafter with three independent sensor systems and upgraded add-on X-by-Wire-System, (**b**) detailed view of a 360° lidar sensor mounted with a FLM connector and (**c**) FLM connector with integrated metal threaded inserts.

Design guidelines already exist for the design of pre-holes in solid thermoplastic components for the insertion of threaded inserts using ultrasonic [19]. The design of the pre-hole contributes significantly to the pull-out behaviour of the threaded insert [20,21]. An undersizing of the hole can lead to stresses and cracks in the thermoplastic component, whereas an oversizing reduces the maximum pull-out forces and excess torques. For the design of the pre-hole, e.g., for the diameter and minimum hole depth, recommendations for different materials are given by the manufacturers in data sheets [22].

For injection moulded components, there are also guidelines on the design of the domes to suit stresses and production, which influence the load-bearing capacity of threaded inserts [23]. Such design guidelines cannot be directly transferred to FLM components, since they do not consider the inhomogeneous material properties and the anisotropic structure in the joining zone. Instead, the need to consider such anisotropy effects can be derived from the material class of fibre-reinforced plastic composites, since they exhibit a comparable anisotropic material configuration. A significant influence of anisotropic material properties on the joining zone quality has previously been demonstrated earlier [8,24].

With regard to the FLM process, variation of the printing process parameters is known to have a high influence on the resulting material properties, as shown by the scientific work of Borowski et al. [25], Bembenek et al. [26], Wang et al. [27] and Kain et al. [28]. It is, therefore, assumed that both the design of the joining zone and the printing parameters have considerable influences on the achievable joint strengths of embedded inserts. However, a detailed investigation on the integration of threaded inserts in FLM components has not been performed thus far.

A systematic understanding of the influence of the joining zone design and selection of printing parameters on the achievable joint properties is also not yet available. This work

focuses on the evaluation of the influence of the joint zone design and the printing process parameters on the pull-out behaviour of warm embedded inserts in FLM test specimens.

## 2. Materials and Methods

### 2.1. Materials and Specimen Manufacturing

To evaluate the load-bearing behaviour of threaded inserts in FLM components, test specimens were manufactured using the FLM process. For this purpose, an Ultimaker 3+ Extended desktop printer was used. The printer was installed in an enclosure to minimize environmental influences.

The tests were performed using specimens made from Extrudr GreenTEC Pro© printing material, which is produced using the biopolymer lignin. Due to the comparatively high mechanical properties shown in Table 1 and application temperatures of up to 160 °C, the material is well-suited for structurally loaded components.

**Table 1.** Material properties of the used printing material GreenTEC Pro© [29].

| Property | Unit | Value |
|---|---|---|
| Tensile Modulus | [MPa] | 4300 |
| Tensile Strength | [MPa] | 58 |
| Elongation at Strength | [%] | 2.8 |
| Melting Temperature | [°C] | 180–200 |
| Density | [g cm$^{-3}$] | 1.35 |

The FLM specimen is a 16 mm high cuboid with a base area of 26 mm × 26 mm. It contains a vertical hole with an 8 mm diameter as pre-hole for the embedding of the insert (Figure 2).

In this paper, the infill density, the wall thickness, the layer height and the nozzle temperature were varied to evaluate their influence on the pull-out behaviour of the inserts. To evaluate the influence of the infill density parameter on the pull-out force of the insert, the FLM test specimens were manufactured with a grid structure and infill densities of 30%, 50% and 70% in the infill area (Figure 2). Increases in infill density resulted in improvements in the tensile strength FLM components [30]. The selected infill densities were chosen to cover the widest possible spectrum while at the same time keeping the manufacturing time as low as possible.

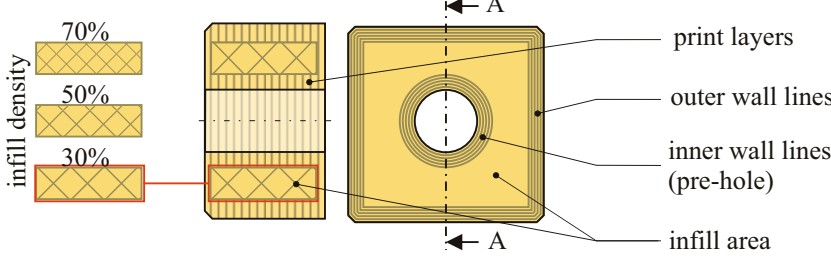

**Figure 2.** Schematic illustration of the FLM test specimen.

The wall thickness, which forms the outer shell of the FLM specimen, is defined by the number of wall lines. From the nozzle diameter of 0.4 mm, the wall thicknesses considered are 1.2 mm (three wall lines), 2.4 mm (six wall lines) and 3.2 mm (eight wall lines). To determine the influence of the layer thickness on the load-bearing capacity of an embedded insert, the FLM test specimens were manufactured with a layer thickness of 0.1 mm, 0.2 mm and 0.3 mm. Increasing the layer height resulted in lowered specimen strengths due to the formation of additional pores.

To avoid this, the layer height should not exceed 80% of the nozzle diameter [31]. Consequently, a maximum layer height of 0.3 mm was chosen for the 0.4 mm nozzle. On

the other hand, the minimum layer height of 0.1 mm results in enhanced print quality while still offering a tolerable print speed. Furthermore, no significant reduction in pore content can be expected below 0.1 mm layer height when using a 0.4 mm nozzle [31]. As six samples were manufactured for each set of parameters, a total of 66 specimens were tested.

For the analysis of the parameters infill density, wall thickness, layer height and nozzle temperature, a constant layer orientation was selected. Two different embedding directions of the insert in correlation to the layer structure were examined in preliminary tests: In one case, the layers were oriented perpendicular to the embedding direction of the insert (Figure 3a; *x–y*-plane). In the second type, the embedding direction of the insert was parallel to the layer structure of the FLM test specimen (Figure 3; *x–z*-plane). As specimens manufactured in the *x–y*-layer orientation exhibited higher pull-out forces and a more predictable failure behaviour, only the specimens with a layer orientation in the *x–y*-plane were investigated in the context of this work.

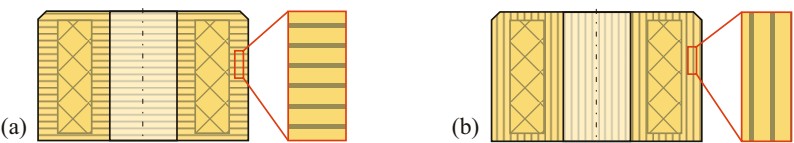

**Figure 3.** Schematic illustration of the orientation of the layer structure in the FLM test specimen: (**a**) layer orientation in the *x–y* plane and (**b**) layer orientation in the *x–z* plane.

The nozzle temperature for printing the specimens was investigated in the process window specified by the manufacturer at the temperatures of 210 °C and 225 °C. Temperatures below 210 °C resulted in high viscosity of the material, which had a negative effect on the material discharge and produced insufficient adhesion to the already deposited material. In contrast, temperatures above 225 °C led to material degradation and material leakage through the nozzle due to decreases of viscosity. Due to the narrow process window, only the minimum 210 °C and maximum 225 °C of the nozzle temperature permissible for production purposes were investigated. A summary of the selected printing process parameters for each varied parameter is shown in Table 2.

**Table 2.** The printing process parameters of the test series for the FLM test specimens.

| Property | Unit | Test Series | | | |
| --- | --- | --- | --- | --- | --- |
| | | Infill Density | Wall Thickness | Layer Height | Nozzle Temperature |
| Bed Temperature | [°C] | 60 | 60 | 60 | 60 |
| Nozzle Diameter | [mm] | 0.4 | 0.4 | 0.4 | 0.4 |
| Count of Top Layers | [-] | 5 | 5 | 5 | 5 |
| Fill Pattern | [-] | Grid | Grid | Grid | Grid |
| Infill Density | [%] | 30–70 | 30 | 30 | 30 |
| Wall Thickness | [mm] | 3.2 | 1.2–3.2 | 3.2 | 3.2 |
| Layer Height | [mm] | 0.2 | 0.2 | 0.1–0.3 | 0.2 |
| Nozzle Temperature | [°C] | 225 | 225 | 225 | 210–225 |

### 2.2. Embedding of Threaded Inserts

For the investigations, Tappex® MULTISERT® BN 37885 threaded inserts without head were used. The brass inserts were integrated in the pre-hole of the FLM specimens via warm embedding (Figure 4a).

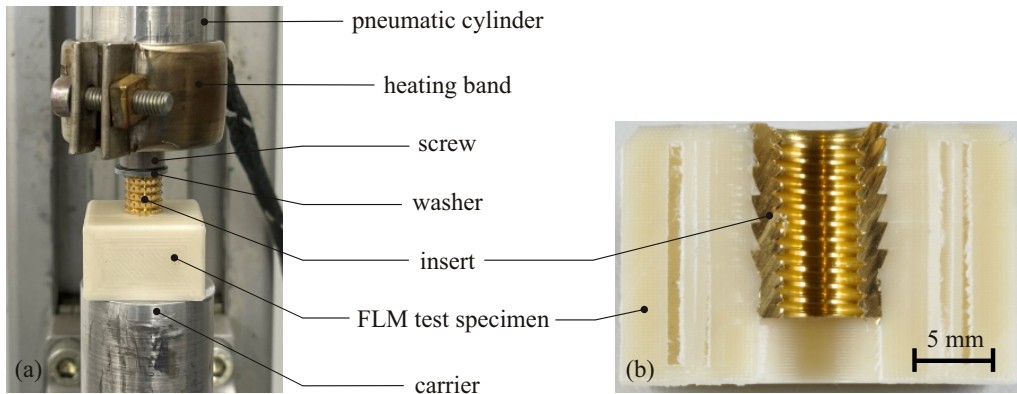

**Figure 4.** (**a**) Set up of the warm embedding of inserts into FLM test specimens and (**b**) cross section of FLM test specimen with embedded insert.

First, an insert is positioned on top of the pre-hole with a thermocouple connected to its surface. The insert is then heated to a temperature of 165 °C. Using the heat transferred from the insert to the FLM specimen, the thermoplastic is plasticized at the contact surface. Then, an axial load of 0.5 kN is applied via a pneumatic cylinder, resulting in the insert being pressed into the now-malleable material and leading to a form-fit joint (Figure 4b).

### 2.3. Test Set-Up

A typical test for evaluating the joint strength of inserts in plastic components is the quasi-static pull-out test. The test setup used in this paper is shown in Figure 5. The test specimen is not clamped, but is pressed against the blank holder by the testing force *F*. The load is applied to the insert by a M6 threaded rod at a constant crosshead velocity of 2 mm/min. The tests were performed using an universal testing machine (Zwick 1465, by ZwickRoell AG, Ulm, Germany).

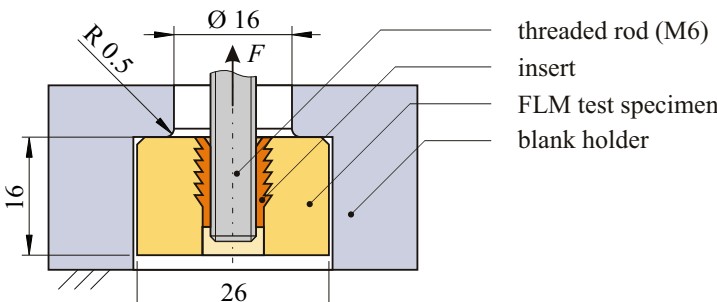

**Figure 5.** Schematic illustration of the pull-out test setup (all dimensions in mm).

## 3. Results and Discussion

### 3.1. Infill Density

In Figure 6, exemplary characteristic force-displacement graphs of the pull-out tests for the variation of the infill density are shown. For each set of parameters, six specimens were tested. The overall characteristics of the graphs are similar, except for the level of the maximum pull-out force and the related displacements.

As can be seen in Figure 7, the pull-out force increases with rising infill density. The test specimens with 30% infill density resulted in an average pull-out force of approximately 1.5 kN and an averaged displacement at maximum pull-out force of 0.6 mm. Increasing the infill density to 50% led to a mean pull-out force of 2.3 kN and an average displacement of 0.7 mm, with a slight increase in the dispersion of the measured values.

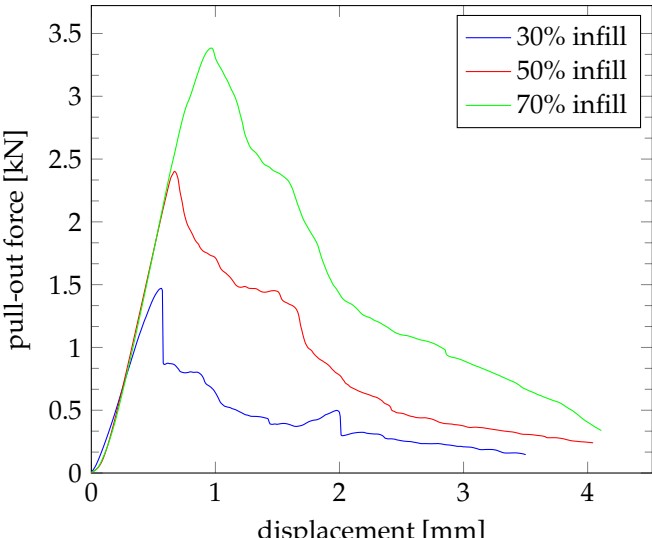

**Figure 6.** Exemplary characteristic force-displacement graphs of the pull-out tests for variations of the infill density.

In the test series with 70% infill, an increase in the mean pull-out force to 3.5 kN and an averaged displacement of 1.2 mm was measured. The dispersion of the maximum pull-out force is on the level of the results with 30% infill density. The results show a significant influence of the increase in infill density with the load-bearing capacity of the joint. Increasing the infill density from 30% to 70% increased the achievable pull-out force by a factor of approximately 2.5. Furthermore, an increase in displacement caused by the increase in filling density was measurable.

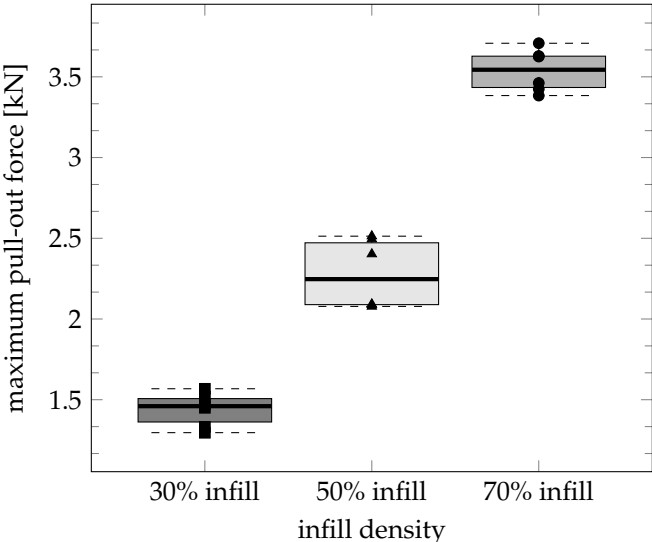

**Figure 7.** Maximum pull-out forces for variations of the infill density (six specimens each).

Figure 8 shows exemplary test specimens of the different test series after the pull-out test. Externally, no difference can be seen between the failure of the test series with increasing infill density. The inserts remained fixed in the wall lines of the pre-hole, which detached from the infill structure due to the force effect. In all test series, interlaminar failure (Figure 8d–f(I)) occurred between the wall lines, particularly in the lower part.

The sectional view in Figure 8f also shows that, with an infill density of 70%, there is a much more pronounced deformation of the wall lines of the pre-hole. Similarly, in all tests, shear failure (Figure 8d–f(II)) occurred between the wall lines and the infill structure.

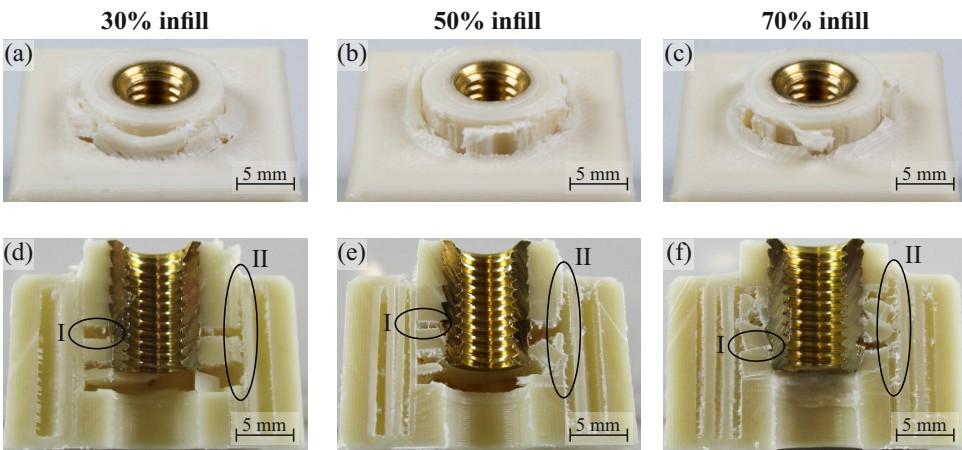

I) interlaminar failure   II) shear failure

**Figure 8.** Exemplary test specimens of the test series with varying infill density: top view (**a**–**c**) and sectional view (**d**–**f**).

The pull-out test applies a tensile force to the insert. The force is transmitted to the wall lines, which are attached in the specimen by the infill structure. This creates a tensile force in the wall lines of the pre-hole and a shear stress in the contact area between the infill structure and the wall lines. By increasing the infill density, the contact area between the wall lines and the infill structure is increased, resulting in higher shear forces that can be absorbed in the interface area between the infill and the wall-line structure. This relation leads to higher tensile forces, which are transmitted into the wall-line structure. These relationships can be seen well by comparing the sectional views in Figure 8d,f. In consideration of the wall-line structure in Figure 8f, more significant deformation can be clearly detected.

The increase in displacement at the maximum pull-out force can be explained by significantly different maximum pull-out forces between the investigated infill densities. Due to the higher forces, more strain is induced into the material, and the elastic range of the material is exploited before abrupt failure occurs between the wall lines of the pre-hole and the infill structure.

### 3.2. Wall Thickness

In addition to the infill density, the wall thickness of the wall lines around the pre-hole in which the insert is embedded was also investigated regarding its influence on the load-bearing capacity. For the evaluation, the maximum pull-out forces and the associated displacements were considered. In Figure 9, exemplary characteristic force-displacement graphs of the pull-out tests are shown. For each set of parameters, six specimens were tested. The overall characteristics of the curves are similar. The force drop of the specimen with three wall lines occurred at a lower level. The displacement at the point of failure was nearly identical for all specimens. As can be seen in Figure 10, enlarging the wall thickness from 1.2 mm (three wall lines) to 2.4 mm (six wall lines) implies a significant increase in the median maximum pull-out force from 1.0 kN to 1.5 kN. The value for the displacement at maximum pull-out force in both test series was 0.47 mm. No significant increase in the value was measured. By enlarging the wall thickness to 3.2 mm (eight wall lines), no increase in the maximum pull-out force and the associated displacement was observed. The dispersion of the measured values is at a similar level for all measurement series. The results show that there is clearly a limit to the minimum wall thickness of the wall lines for the type of insert investigated.

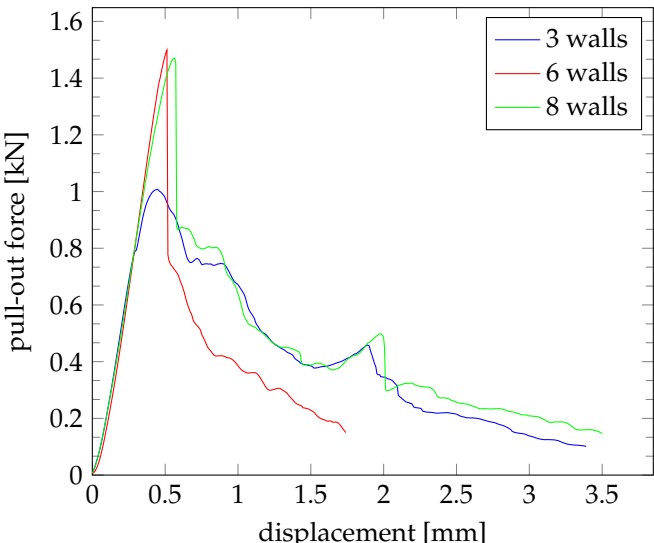

**Figure 9.** Characteristic force-displacement graphs of the pull-out tests for variations of the wall thickness.

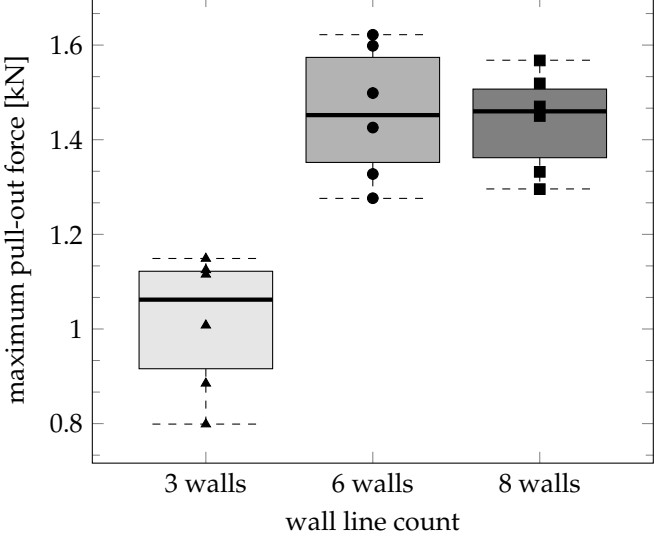

**Figure 10.** Maximum pull-out forces for variations of the wall thickness (six specimens each).

An external examination of the test specimens shows that the wall lines of the pre-hole are detached from the infill structure in all test series (Figure 11a–c). The sectional views show an interlaminar failure of the wall structure equivalent to the other parameters investigated. Both the top views (Figure 11a–c) and the sectional views (Figure 11d–f) demonstrate that the inserts of the test series with six and eight wall lines, respectively, remain fixed in the wall lines of the pre-hole. However, Figure 11d shows a detachment of the insert from the wall lines.

The combined consideration of the achieved maximum pull-out forces and the fracture figures leads to the assumption that wall thicknesses that are too small form a lower anchorage, leading to a detachment of the insert. Thus, a lower load-bearing capacity of the joint is achieved. Another aspect explaining the lower maximum pull-out forces with smaller wall thickness is the shell surface at the contact area between the wall lines of the pre-hole and the infill structure. By increasing the number of wall lines, the shell area is increased and, thus, also the contact area of the infill structure. The increase of the shell surface results in higher shear forces that can be absorbed in the interface area between the the infill and the wall-line structure. This relation leads to higher tensile forces, which

are transmitted into the wall-line structure. This was observed for increasing the wall line number from three to six. However, when comparing specimens with six and eight wall lines, there seems to be a limit to the maximum pull-out forces, because the failure behaviour is dominated by the low infill density of 30%.

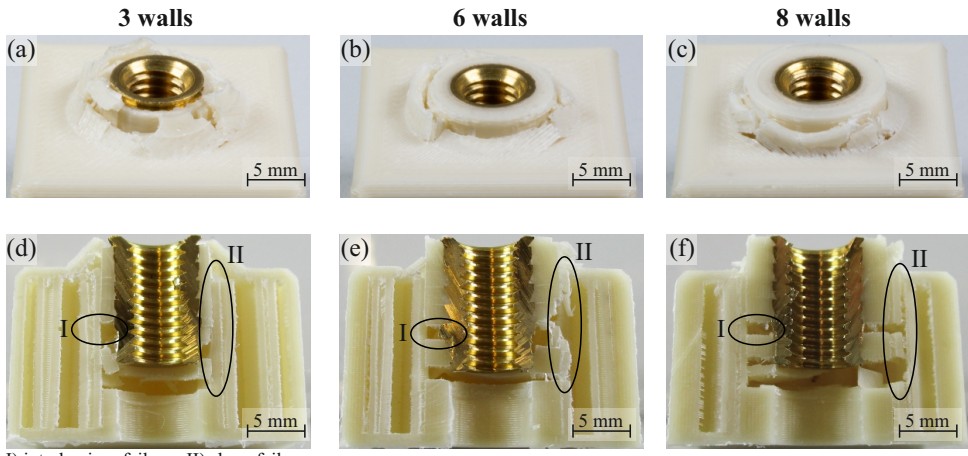

**Figure 11.** Exemplary test specimens of the test series with varying wall thickness and corresponding sectional views: top view (**a**–**c**) and sectional view (**d**–**f**).

In contrast to the results of varying the infill density, the mean displacements at the point of failure are nearly the same for all wall thicknesses. Therefore, it can be concluded that an increase of the displacement at maximum pull-out force appears only at higher forces. An enhancement of the maximum pull-out forces of 0.5 kN does not influence the associated displacement significantly.

### 3.3. Layer Height

The layer height is an elementary parameter for the resolution of AM components and, therefore, a quality factor that influences the component properties. Consequently, the influence of the layer height on the load-bearing capacity of the embedded insert was investigated. Exemplary characteristic force-displacement graphs of the pull-out tests for variation of the layer height are shown in Figure 12.

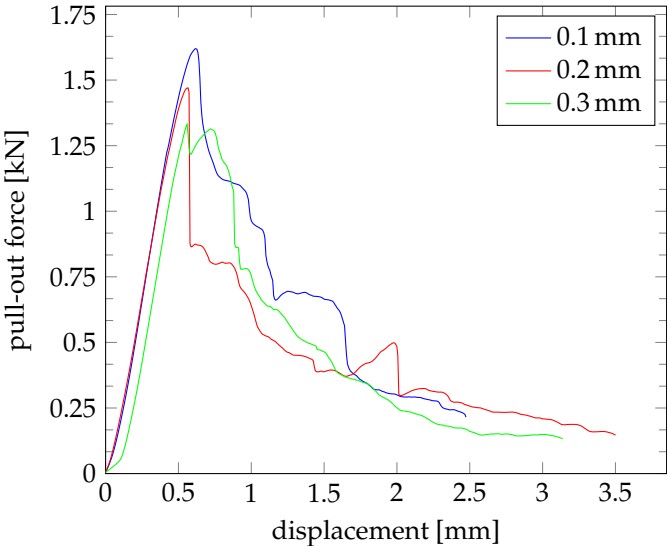

**Figure 12.** Characteristic force-displacement graphs of the pull-out tests for variations of the layer height.

For each set of parameters, six specimens were tested. The characteristics of the graphs are almost congruent. The diagram in Figure 13 shows the maximum pull-out forces.

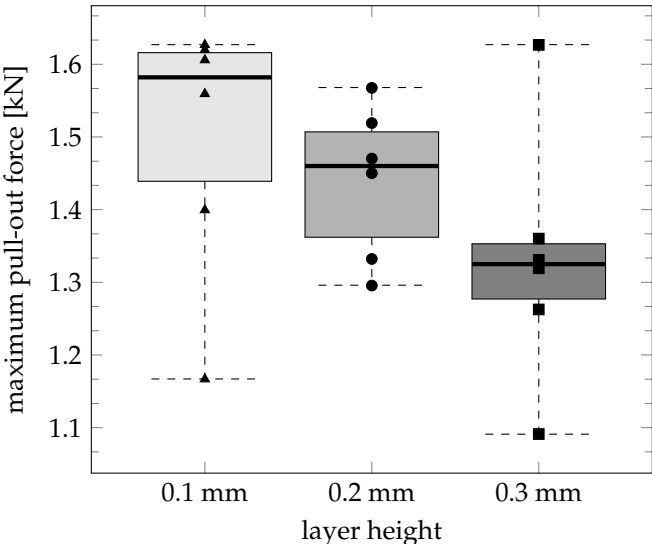

**Figure 13.** Maximum pull-out forces for variations of the layer height (six specimens each).

The maximum pull-out force decreased with increasing layer height. At a layer height of 0.1 mm, the median maximum pull-out force was approximately 1.6 kN, and the associated averaged displacement was 0.6 mm. The dispersion of the measured values is the largest in comparison with the other considered layer heights—in particular, the minimum value shows a large deviation from the median. The test specimens produced with a layer height of 0.2 mm achieved a median pull-out force of is approximately 1.5 kN and an averaged displacement of 0.6 mm with a similarly large interquartile range. However, the deviation of the maximum and minimum values from the median were significantly smaller. Increasing the layer height to 0.3 mm caused a reduction in the median maximum pull-out force to 1.3 kN and an associated averaged displacement of 0.6 mm. Both the maximum and minimum deviations were clearly pronounced, with amounts of approximately 300 N.

The sectional views of the test series with varied layer height (Figure 14d–f) show a detachment of the wall lines of the pre-hole from the infill structure in all specimens. The tested specimens with a layer height of 0.1 mm (Figure 14d(I)) and 0.2 mm (Figure 14e(I)) are characterized by a pronounced delamination between the wall lines of the pre-hole.

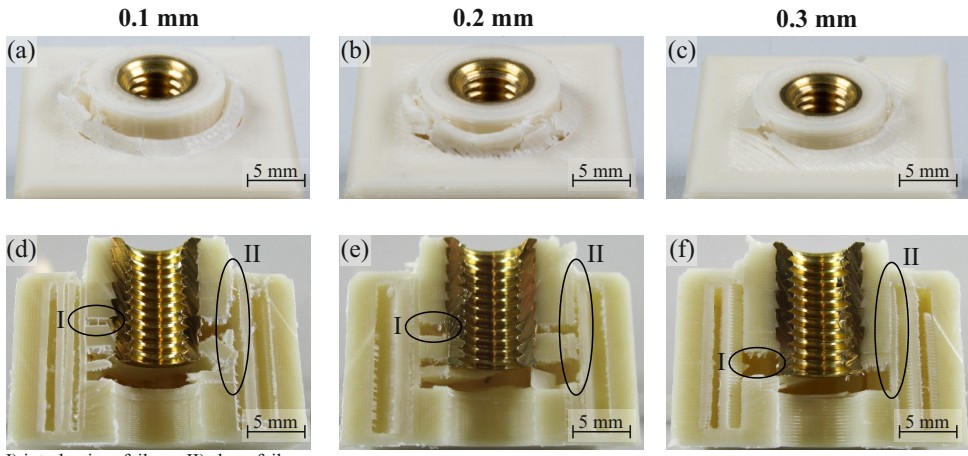

**Figure 14.** Exemplary test specimens of the test series with varying layer heights: top view (**a**–**c**) and sectional view (**d**–**f**).

The specimen with a layer height of 0.3 mm (Figure 14f(I)) shows decreased delaminations of the wall lines within the joining area of the insert. The higher pull-out forces in combination with the reduced delamination phenomena indicate better bonding between the wall lines of the pre-hole and the infill structure at lower layer height.

As in the investigations of the wall thickness, there were only minor deviations in the displacement between the test series. As of the small deviation of the maximum forces of 300 N between the series of measurements of the investigated parameters, no significant change of the displacement at maximum pull-out force was detectable.

### 3.4. Nozzle Temperature

In Figure 15, exemplary characteristic force-displacement graphs of the pull-out tests for the variation of the nozzle temperature are demonstrated.

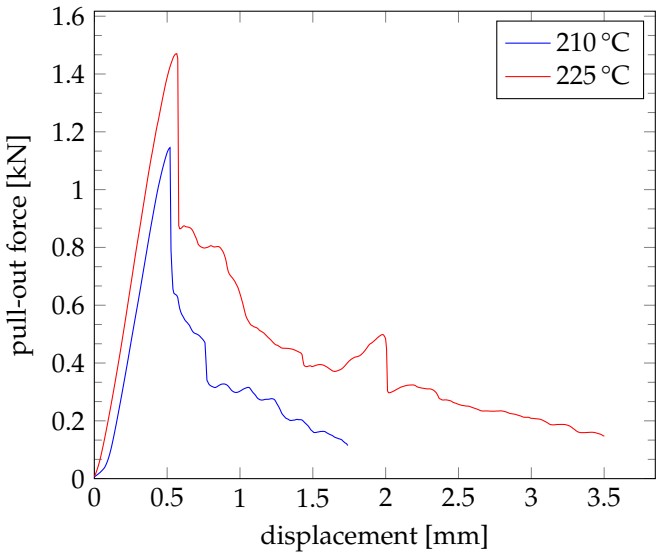

**Figure 15.** Characteristic force-displacement graphs of the pull-out tests for variations of the temperature.

For each parameter, six specimens were tested. The overall characteristics of the graphs are similar except for the level of the maximum pull-out force and the related displacements. The maximum pull-out forces are compared in Figure 16.

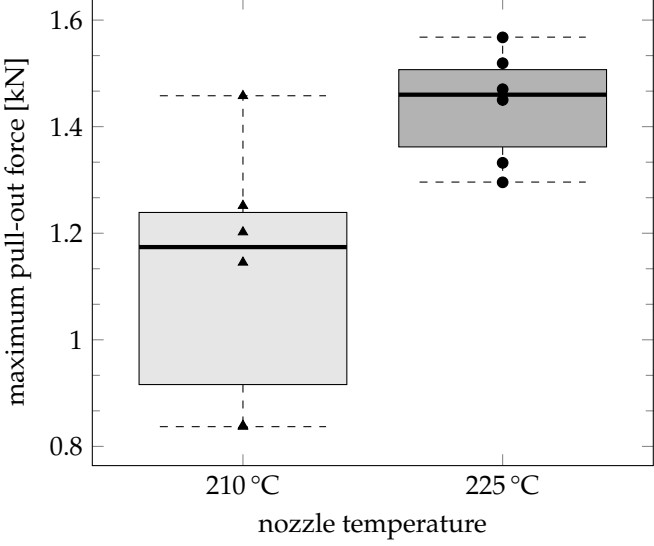

**Figure 16.** Maximum pull-out forces for variations of the nozzle temperature (six specimens each).

Reducing the nozzle temperature from 225 °C to 210 °C led to a decrease of the median maximum pull-out force from approximately 1.5 kN to approximately 1.2 kN. The associated mean displacement values were 0.6 mm and 0.5 mm. In addition, by reducing the nozzle temperature, a significant increase in the deviation of the results was observed. The external view of the tested specimens (Figure 17a,b) shows no significant differences in the failure mode. However, the sectional view in Figure 17c clearly demonstrates that there is no interlaminar failure of the layers in the wall lines of the pre-hole; there is only a shear failure in the shell surface, and the structure is pulled out of the specimen as a monolithic body. In contrast, the specimen printed at 225 °C (Figure 17d(I)) is characterised by interlaminar failure between the layers of the wall lines of the pre-hole. The failure modes indicate that the bonding between the infill structure and the wall lines of the pre-hole was weakened by the lower nozzle temperature. The interlayer bonding between the layers of the wall lines of the pre-hole appears to be stronger than the bonding between the infill structure and the wall lines of the pre-hole at the selected printing parameters.

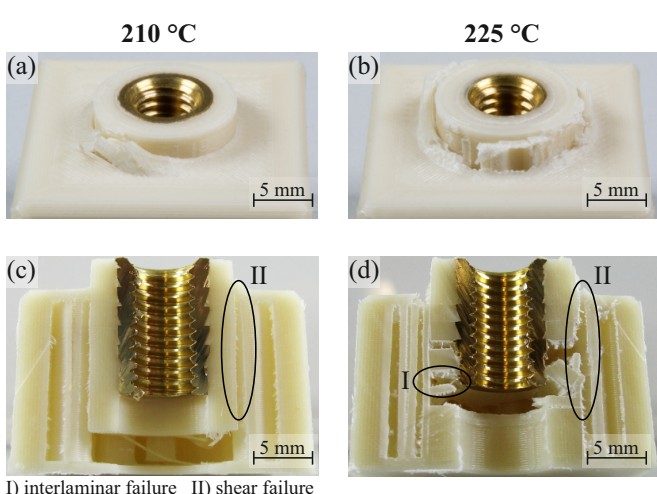

I) interlaminar failure    II) shear failure

**Figure 17.** Exemplary test specimens of the test series with varying nozzle temperature: top view (**a**,**b**) and sectional view (**c**,**d**).

There are only minor variations in the displacement between the test series, similar to the layer height investigations. No appreciable change in the displacement is discernible due to the small variation in the 300 N maximum forces between the series of measurements of the investigated parameters.

## 4. Conclusions and Outlook

Metal threaded inserts are already used in AM components as extrinsic interfaces to attach them to other parts or assemblies. Warm embedding is an established integration method for inserts in thermoplastic components. In order to evaluate the influence of the joining zone design and the printing process parameters on the achievable joint properties, warm-embedded inserts in FLM test specimens were investigated in pull-out tests. We demonstrated that a higher infill density led to an increase in joint strength. An examination of the number of wall lines around the pre-hole in which the insert was embedded showed that there was a minimum value of wall lines for the selected insert geometry to achieve a high pull-out force.

Maximizing the number of wall lines beyond this did not significantly affect the joint strength. Furthermore, for the type of insert investigated, the strength of the joint was improved by reducing the layer height. In addition to the design of the joining zone, the printing parameters also have an influence on the pull-out behaviour, which was demonstrated by the example of the nozzle temperature. The infill density had the largest impact on the joint strength. The holistic view of the investigation resulted in the following

recommendations for the design of a strong joint for the considered type of insert and printing material:

- the infill density should be 70% or higher;
- the minimum wall thickness should be 2.4 mm;
- the layer height should be 0.1 mm or smaller;
- the printing temperature should be 225 °C.

In future investigations, the influence and correlation of the insert geometry, the embedding method and further printing parameters should be considered. Furthermore, an analysis of different printing materials on the joint strength could be of interest as well as investigations of further load types.

**Author Contributions:** Conceptualization, T.K. and J.T.; methodology, J.T.; investigation, T.K.; writing—original draft preparation, T.K., J.T. and C.V.; writing—review and editing, T.K., J.T., T.B., M.G. and N.M.; visualization, T.K., J.T. and C.V.; supervision, M.G. and N.M. All authors have read and agreed to the published version of the manuscript.

**Funding:** This research received no external funding.

**Data Availability Statement:** The data presented in this study are available on request from the corresponding author.

**Acknowledgments:** The authors gratefully acknowledge the assistance of Sven Oldewurtel in manufacturing the test specimens and conducting the investigations as part of their student work at the Institute of Lightweight Engineering and Polymer Technology, Technische Universität Dresden.

**Conflicts of Interest:** The authors declare no conflict of interest.

## Abbreviations

The following abbreviations are used in this manuscript:

| | |
|---|---|
| AM | Additive Manufacturing |
| FLM | Fused-Layer Modelling |

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
