# Peer review of "Investigation of the Pull-Out Behaviour of Metal Threaded Inserts in Thermoplastic Fused-Layer Modelling (FLM) Components"

_jmmp, doi:10.3390/jmmp7010042_

Round 1

Author Response

Thank you for the comments on our article. We have edited all your suggestions and transferred them to a new manuscript. Please see the attachment.

Reviewer 2 Report

This study has attempted to investigate threaded inserts in additively manufactured thermoplastic components. It seems to be a timely and appropriate study since the outcome can have a positive contribution to industrial applications. The paper is well-organized, contains original results and deserves to be published after major revision. The English of the paper should be significantly improved.

v Abstract

ü  It is highly recommended to follow the protocol for writing a standard “abstract”. One sentence to state the importance of the issue, followed by the explicit illustration of the aims, employed methods, main achievements, and an outlook in the light of obtained results. Please rewrite the section in the light of the mentioned protocol. Moreover, the authors are recommended to clearly address the affecting operating factors in this work.  

ü  The authors are recommended to add some quantitative data in this section, reporting the increase/decrease (%) in a property by a change in operating factor. 

v Introduction

ü  This section should be strengthened by more references. The following papers should be cited in the paper:

·       doi.org/10.1016/j.polymer.2017.03.011

·       doi.org/10.3390/jmmp6030065

·       doi.org/10.1007/s00107-019-01473-0

ü  In the final paragraph of the “Introduction” the authors have stated the main gaps in the literature; However, it is better to clearly address how they are going to fill these gaps in this study.

v Materials and methods

ü  The authors studied the influence of layer thickness in the range of 0.1-0.3 mm. Why haven’t they chosen thicker or thinner thicknesses? Same comment for the selection of infill density in the range of 30-70%.  

ü There is no sentence in the main text on what Table 1 shows. Please add it to the text. The mere Table caption is not enough.  

ü  Line #102, it should read “heated to a temperature of 165 °C.”

v Results   

ü  Line # 118, please use “overall characteristics” rather than “principle characteristics”.

ü  Please add numerical values of the obtained graphs in Fig. 5, namely displacements, to the text.

ü  Please use the same unit for the vertical axis in all figures. They should be either N or kN.

ü  Line #128, it is better to use “Increasing” instead of “Heightening”.

ü  Please provide scientific reasons why the increased infill density has resulted in higher pull-out force.

ü  In Fig. 7, it is recommended to highlight failure features, such as interlaminar failure, by the arrows.    

ü  SEM analysis of the fractured surface can provide fruitful information on the fracture type. It is optional but will enhance the quality of the paper. 

ü  Similar to my previous comments on the “Infill density” section, in the next sections dealing with wall thickness, layer height, etc., a detailed discussion with stressing mechanisms should be presented.

v Conclusions

ü  In this section, the optimum processing condition in terms of Infill density, wall thickness, layer height, etc. should be explicitly stated.

ü  Addressing potential application(s) can attract much more attention to the paper.

Author Response

(The authors gave the same response as above.)

Reviewer 3 Report

There is very interesting paper, and research methodology were prepared properly. In my opinion, in the introduction section, the real application possibilities should be underlined. I have a few methodological and editorial suggestions and questions:

1) "Infill area" and "Infill density" - what does it mean? There is a lack of clear definition (with scheme or something like a dimension analysis or math description), so those terms are not clear for the audience.

2) What is the reason of the differences in the displacement, when the maximum forces are gained for different infill density (Fig. 5)? Please explain.

3) To obtain uniformity of the research methodology, for the last test (different temperatures) the 3 different temperatures should be analyzed.

4) There is lack of information about the rest parameters for the particular tests (for example, there is a lack of information about temperature, number of layers and nozzle diameter during analyzing the influence of infill density - I understand that the rest parameters were constant, but there should be information about their values). In the ideal situation, all the values which are analyzed should be considered together - this activity would give 243 combinations (3 temperature, 3 sizes of nozzles, 3 numbering of walls and 3 infill densities). I understand, that this activity can be considered for different publication (with using for example multiple-variant analysis), but in such a case there should be information about it in conclusions section.

5) In my opinion, the conclusions can be enriched by a comment about influence of the observations to a practical approach connected with manufacturing or utilizing such a connection.

6) There is lack of information about materials - from what material the insert is made? What kind of plastic is under market name "Extrudr GreenTEC Pro"? What about its mechanical properties? 

Author Response

(The authors gave the same response as above.)

Reviewer 4 Report

Dear Authors,

The idea of combining metal inserts with a printed housing, presented by you, is in line with the current trends in the development of manufacturing.

Despite the interesting topic, the presented article requires significant corrections:

Please specify the basic mechanical properties of the Extrudr GreenTEC Pro© material.

The process of maintaining the temperature by the nozzle changes between 210 and 225 degrees Celsius. The process of pressing the sleeve into the plastic housing also introduces many variables. In this case, it is required to carry out at least 3 tensile tests for each variant (if the results are reproducible, then a minimum of 5 tests.

A tensile test should also be performed on a reference item to be able to compare the results obtained.

Authors should test the limit torque specified by the manufacturer for reference bushings. This information will determine whether we are able to properly tighten the threaded element.

Author Response

(The authors gave the same response as above.)

Round 2

Reviewer 2 Report

The authors have fully applied my comments. From my side, the present version of the manuscript deserves to be published in “JMMP”.  

Reviewer 4 Report

Dear authors,

I accept the changes made